# Investigation of Intestinal Microbes of Five Zokor Species Based on 16S rRNA Sequences

**DOI:** 10.3390/microorganisms13010027

**Published:** 2024-12-26

**Authors:** Yao Zou, Quan Zou, Hui Yang, Chongxuan Han

**Affiliations:** 1Yangtze Delta Region Institute (Quzhou), University of Electronic Science and Technology of China, Quzhou 324000, China; zy1179520342@163.com (Y.Z.); zouquan@nclab.net (Q.Z.); 2Key Laboratory of National Forestry and Grassland Administration on Management of Western Forest Bio-Disaster, Northwest Agriculture and Forestry University, Yangling 712100, China; 3Institute of Fundamental and Frontier Sciences, University of Electronic Science and Technology of China, Chengdu 610054, China

**Keywords:** subterranean rodents, intestinal microbes, zokor, herbivores, metabolism

## Abstract

Zokor is a group of subterranean rodents that are adapted to underground life and feed on plant roots. Here, we investigated the intestinal microbes of five zokor species (*Eospalax cansus*, *Eospalax rothschildi*, *Eospalax smithi*, *Myospalax aspalax*, and *Myospalax psilurus*) using 16S amplicon technology combined with bioinformatics. Microbial composition analysis showed similar intestinal microbes but different proportions among five zokor species, and their dominant bacteria corresponded to those of herbivores. To visualize the relationships among samples, PCoA and PERMANOVA tests showed that the intestinal microbes of zokors are largely clustered by host species, but less so by genetics and geographical location. To find microbes that differ among species, LefSe analysis identified *Lactobacillus*, *Muribaculaceae*, *Lachnospiraceae_NK4A136_group*, *unclassified_f_Christensenellaceae*, and *Desulfovibrio* as biomarkers for *E. cansus*, *E. rothschildi*, *E. smithi*, *M. aspalax,* and *M. psilurus*, respectively. PICRUSt metagenome predictions revealed enriched microbial genes for carbohydrate and amino acid metabolism in *E. cansus* and *E. smithi*, and for cofactor and vitamin metabolism as well as glycan biosynthesis and metabolism in *E. rothschildi*, *M. aspalax*, and *M. psilurus*. Our results demonstrated differences in the microbial composition and functions among five zokor species, potentially related to host genetics, and host ecology including dietary habits and habitat environment. These works would provide new insight into understanding how subterranean zokors adapt to their habitats by regulating intestinal microbes.

## 1. Introduction

In recent years, the study of intestinal microbes in human and animal has received extensive attention [1,2,3]. The mammalian intestinal microbiota, a complex and diverse community, encodes 150 times more genes than the host and affects multiple aspects of host biology through host–microbe interactions [4]. Hosts help form microbial communities by actively eliminating disease-causing microbes while allowing beneficial ones to remain, or tolerating harmless ones [5]. Intestinal microbes are involved in the decomposition of complex carbohydrate. By compensating for the digestive capacity of the host, they play a crucial role in intestinal nutrition [6]. They are closely related to host digestion, metabolism, immunity, and health [7,8,9]. Therefore, exploring host–microbe interactions could help us better understand the ecology and evolution of wild animals, and is of great significance for understanding host energy metabolism and its ecological adaptability [10].

The zokor (Myospalacinae) is a typical subterranean rodent distributed in East Asia. It lives underground all year round, digging tunnels to obtain plant roots for food [11]. Their underground lifestyle and tunnel environment give them characteristics adapted to underground life, distinct from aboveground counterparts: visual degradation, strong forelimbs and developed skeletal muscles [12], developed cecum [13], and high hemoglobin level [14]. Studies show that subterranean rodents conduct all their activities within their tunnels. Meanwhile, building and maintaining the tunnel system is an extremely energy-intensive process [15,16]. Zokors also engage in energy-intensive digging to obtain and store food. Additionally, food serves as the primary source of energy for zokors. However, due to their underground lifestyle, the food options for zokors are extremely restricted. As a result, they frequently encounter a shortage of food resources and poor food quality [17]. The elevated energy demand of zokors calls for special digestive and metabolic strategies. Meanwhile, diverse intestinal microbes might be an adaptation to these particular strategies. As herbivorous rodents, zokors have intestinal microbes that are particularly important for complex carbohydrates that cannot be degraded by the host. Previous studies have shown that Firmicutes and Bacteroidetes were the two dominant phyla in intestinal microbes of zokors [18,19,20,21,22,23,24]. At the genus level, zokors rely on intestinal microbes for fermentation, which makes them rich in microbes associated with fiber fermentation, such as *Lachnospiraceae_NK4A136_group*, *Ruminocuccus*, and *unclassified_f_Lachnospiraceae* [18,19,24]. Furthermore, some bacteria, such as *norank_f_Muribaculaceae*, *Lactobacillus*, and *Desulfovibrio* are also enriched in the intestine of some zokor species [18,22,23,24]. At present, there is a lack of comparative studies on the intestinal microbes of multiple zokor species.

Growing evidence suggests that changes in gut microbes throughout life are triggered by a variety of factors, such as host genotype [25], diet [26,27], hormone levels [28], age, lifestyle, social networks, and host ecology [29,30,31]. It remains unknown which factors affect the intestinal microbes of zokors and how these microbes vary among zokor species. In this study, we investigated the intestinal microbial composition of five wild zokor species by analyzing 16S rRNA sequences. Additionally, we screened biomarkers to explore the adaptability of zokors to special food and environment through intestinal microbes. Investigating the intestinal microbes is beneficial for exploring the nutritional requirements of zokors and further understanding their habitat and feeding selection strategies. These efforts would offer theoretical references for further formulating an ecological control strategy for rodent damage.

## 2. Materials and Methods

### 2.1. Sample Collection

Wild adult zokor samples were collected during the period from 2019 to 2021. *E. cansus* (EC, *n* = 45) and *E. smithi* (ES, *n* = 11) were captured from Ningxia Hui Autonomous Region of China, and *E. rothschildi* (ER, *n* = 20) were from Shaanxi Province of China. The sample information was shown in Appendix A. Underground trapping method was used to trap all the zokors. After the zokors were sedated with xylazine hydrochloride (5 mg/kg), they were humanely euthanized by intravenous overdose of pentobarbital sodium (390 mg/mL). The cecal contents were then placed into 5 mL sterilized storage tubes, immediately stored in liquid nitrogen, and then stored in a laboratory freezer at −80 °C.

### 2.2. DNA Extraction, PCR Amplification and Amplicon Sequencing

Total cecal genomic DNA was extracted from the cecum contents using the Stool Genomic DNA Extraction Kit (TianGen, Beijing, China) following the manufacturer’s protocol. The DNA was analyzed for purity using 1% agarose gel electrophoresis; the DNA concentration was accurately quantified using a Nanodrop 2000 Spectrophotometer (Thermo Fisher Scientific, Wilmington, NC, USA). Then, it was sent to Majorbio Bio-Pharm Technology Co., Ltd. (Shanghai, China) for purification and sequencing.

The hypervariable regions V3-V4 of the bacterial 16S rRNA were amplified with primers (338F: 5′-ACTCCTACGGGAGGCAGCAG-3′, 806R: 5′-GGACTACHVGGGTWTCTAAT-3′) [32]. PCR reactions were performed in a 25 μL reaction volume consisting of a 1.0 μL template DNA, 1.0 μL forward and reverse primers (10 μmol·L^−1^), 0.5 μL dNTPs mix (10 mM concentration of each dNTP), 5 × High GC Buffer 5 μL, and 0.25 μL Q5 high-fidelity DNA polymerase, with sterile distilled water added up to 25 μL volume. The PCR reaction was carried out under the following conditions: (1) initial denaturation at 95 °C for 3 min; (2) 27 cycles of denaturation at 95 °C for 30 s; (3) annealing at 55 °C for 30 s; (4) elongation at 72 °C for 45 s; and (5) final extension 72 °C for 10 min. The PCR productions were quantified using QuantiFlourTM-ST (Promega, Madison, WI, USA) in accordance with the manufacturer’s protocols.

Paired-end sequencing (2 × 300) was performed on the Illumina MiSeq platform (Illumina, San Diego, CA, USA) by Majorbio Bio-Pharm Technology Co., Ltd. (Shanghai, China).

### 2.3. Data Collation, Analyses, and Visualization

Raw reads of the V3-V4 hypervariable regions of the cecal 16S rRNA of three zokor species from *Eospalax* were obtained based on sequencing methods. In addition, raw sequences of V3-V4 hypervariable regions of the cecal 16S rRNA of *Myospalax aspalax* (MA, *n* = 15) and *Myospalax psilurus* (MP, *n* = 18) were downloaded from the NCBI Sequence Read Archive under BioProject PRJNA1089538 [19]. All the raw sequences were processed using the QIIME2 pipeline (version 2023.5) [33]. A DADA2 plugin in the QIIME2 pipeline was employed to filter sequences, trim primers, eliminate chimeric reads, and merge paired-end reads [34]. Merged reads were clustered into amplicon sequence variants (ASVs) based on sequence similarity. To standardize the sampling work among samples, the feature-table plugin was used to rarefy each sample to an equal depth, and the abundance table of each sample at the ASV level was finally obtained. The phylogenetic tree based on ASV representative sequences was constructed using FastTree [35]. In addition, reference sequences and corresponding taxonomy were downloaded based on the SILVA (v138) database [36]. After the reference sequence was extracted to the V3-V4 hypervariable regions, the Naive Bayes method was used to train classifier based on reference sequences and corresponding taxonomic information. A trained classifier can annotate ASVs for classification. Reads identified as archaea or chloroplast were removed. Next, the alpha diversity (Shannon index [37], Chao index [38]) and beta diversity were estimated based on ASV-table and phylogenetic tree. R v4.3.1 software was used to visualize the results [39].

All the raw sequencing data have been deposited in the NCBI Sequence Read Archive under the BioProject PRJNA664217, PRJNA833598 and PRJNA1130553.

### 2.4. Statistical Analyses

Non-parametric Wilcoxon rank sum tests were performed to assess the differences in the alpha diversity indexes across the pairwise rodent species [40]. The structure of the intestinal microbial communities among the species were compared by computing dissimilarity values between pairwise samples based on Bray–Curtis [41], and unweighted UniFrac metrics [42]. Furthermore, principal coordinate analysis (PCoA) plots and non-metric multidimensional scaling (NMDS) analysis plots were constructed to visualize the relationships among the samples [43]. To estimate the overall variance contribution of the host species to microbes, a permutational analysis of variance (PERMANOVA, permutations = 999) was conducted by utilizing the adonis2 function in the vegan R package based on Bray–Curtis distances [39,44].

Linear discriminant analysis effect size (LefSe) was performed to find taxa with significant differences among the groups [45]. The threshold of LDA score was set to be greater than 4 and was utilized to draw histograms of enriched taxa for each zokor species.

PICRUSt2 analysis was used to predict the microbial functions of genes based on ASV-table and representative sequences [46]. We compared the relative abundances of the gene function among zokor species based on the KEGG database [47]. Anova tests were performed to compare the differences in predicted abundance among five zokor species.

### 2.5. Ethical Statement

All the animal experiments were approved by the Ethics Committee of University of Electronic Science and Technology of China (Approval Code: 106142024103031567; Approval Date: 30 October 2024). The processing of wild animals and sample collection were strictly congruent with the guidelines of our academic institution.

## 3. Results

### 3.1. Intestinal Bacterial Composition Among Rodents from Two Families

A total of 6212687 raw paired-end reads were obtained from 109 samples. After the DADA2 analysis pipeline, these sequences were grouped into 10206 ASVs that belonged to 14 phyla, 238 genera.

We analyzed the intestinal microbial composition of five zokor species under the phylum and genus levels. The 14 microbial phyla with maximum relative abundance of each zokor species are shown in Figure 1a. At the phylum level, sequences originating from Firmicutes (63.20%) and Bacteroidota (29.65%) were dominant, followed by Desulfobacterota (4.69%) and Proteobacteria (1.00%) (>1% relative abundance). However, there were some differences in the microbial proportion among the five species (Figure 1a). The proportion of Firmicutes was 71.39%, 44.59%, 85.47%, 52.82%, 58.44%, while Bacteroidota was 21.60%, 44.96%, 12.48%, 41.22%, 33.63% in *E. cansus*, *E. rothschildi*, *E. smithi*, *M. aspalax*, and *M. psilurus*, respectively. Firmicutes were relatively more abundant in *E. cansus* and *E. smithi*, while Bacteroidota were more abundant in *E. rothschildi*, *M. aspalax*, and *M. psilurus*.

The top 30 microbial genera with maximum relative abundance of each zokor species are shown in Figure 1b. At the genus level, sequences originating from Muribaculaceae (28.58%) were dominant, followed by *Lachnospiraceae_NK4A136_group* (10.37%), *unclassified_f_Lachnospiraceae* (9.74%), *Desulfovibrio* (4.67%), *uncultured_f_Lachnospiraceae* (4.52%), *uncultured_f_Oscillospiraceae* (4.29%), *uncultured_f_Christensenellaceae* (3.58%), *Ruminococcus* (3.23%), *unclassified_f_Oscillospiraceae* (2.55%), *Lactobacillus* (2.28%) and *Eubacterium_siraeum_group* (2.24%) (>2% relative abundance). However, some microbial genera showed significantly differences among the five zokor species (Appendix A). *Muribaculaceae* were more abundant in *E. rothschildi*, and *M. aspalax*, while *Lachnospiraceae_NK4A136_group* and *unclassified_f_Lachnospiraceae* were more abundant in *E. smithi*. Furthermore, *Desulfovibrio*, *uncultured_f_Lachnospiraceae* and *uncultured_f_Oscillospiraceae* were more abundant in *M. psilurus*, while *Ruminococcus* were abundance in *E. cansus* and *E. smithi* (Appendix A).

### 3.2. Intestinal Bacterial Diversity Among Rodent Species

We estimated the alpha diversity by using the Chao index (Figure 2a) and Shannon index (Figure 2b). We also compared the differences in alpha diversity among zokor species. It was indicated that *E. cansus* had a significantly higher Chao index and Shannon index than those species from *Myospalas*, although there were differences within the genera. In *Eospalax*, *E. cansus* had significant higher alpha diversity than that of *E. smithi*. Moreover, although *E. cansus* had a higher alpha diversity than that of *E. rothschildi*, the results showed no statistical significance.

PCoA analysis is shown in Figure 3. The intestinal microbes of rodent species mainly clustered by host species are based on Bray–Curtis distance matrices (Figure 3a). However, PCoA analysis based on unweighted UniFrac matrices revealed that the intestinal microbes of the zokor species were mainly clustered by host genera. Specifically, *E. cansus*, *E. rothschildi*, and *E. smithi* from *Eospalax* clustered together, while *M. aspalax* and *M. psilurus* from *Myospalax* were more closely clustered (Figure 3b).

PERMANOVA tests based on Bray–Curtis distance showed that host species significantly explained 12.15% (F = 3.60, R^2^ = 0.1215, *p* = 0.001) of the beta diversity among different samples. This explanation was higher than that explained by genera from which the host comes (F = 6.74, R^2^ = 0.0593, *p* = 0.001) and geographical locations (F = 5.35, R^2^ = 0.0916, *p* = 0.001). NMDS analysis based on Bray–Curtis (Appendix A, stress = 0.1535, *p* < 0.01) showed that the intestinal microbes of five zokor species had a distinct structure.

### 3.3. Differences Taxa Among Five Zokor Species

LefSe analysis was conducted to screen the biomarkers that differ among zokor species, which revealed taxa with significant differences among five zokor species (Figure 4 and Appendix A, LDA score > 4.0). We found that 4, 11, 8, 1, and 8 bacterial taxa (LDA > 4.0) were enriched in *E. cansus*, *E. rothschildi*, *E. smithi*, *M. aspalax*, and *M. psilurus*, respectively (Figure 4). At the phylum level, Firmicutes, Bacteroidota, and Desulfobacterota were the biomarkers of *E. smithi*, *E. rothschildi,* and *M. psilurus*, respectively. At the genus level, *Lactobacillus*, and *unclassified_f_Christensenellaceae* were the biomarkers of *E. cansus* and *M. aspalax*, respectively. Furthermore, *Muribaculaceae*, *uncultured_f_Christensenellaceae*, and *Eubacterium_brachy_group* were the biomarkers of *E. rothschildi*. *Lachnospiraceae_NK4A136_group*, *unclassified_f_Lachnospiraceae*, and *Ruminococcus* were the biomarker of *E. smithi*. *Desulfovibrio*, *uncultured_f_Lachnospiraceae*, *unclassified_f_Oscillospiraceae*, *uncultured_f_Oscillospiraceae* were the biomarkers of *M. psilurus*.

### 3.4. Predicted Metagenomes Among Zokor Species

Predicted metagenomes based on KEGG database showed that the microbial gene functions in all the samples were predominantly in metabolism (70.61%) and genetic information processing (12.11%) (Appendix A). Under the metabolism function of KEEG level 1, there were 13 predicted gene functions in all the samples at the KEGG level 2. Among them, 10 predicted gene functions exhibited significant differences among the zokor species, including amino acid metabolism and carbohydrate metabolism (Figure 5 and Appendix A). These two functions had the highest abundance in all the samples (Figure 5).

## 4. Discussion

Intestinal microbes play a crucial role in nutrient absorption, metabolism, and immunity of the host [48,49]. Elucidating the composition and function of intestinal microbes in rodents is highly significant for exploring the ecology, evolution, and ecological adaptability of wild rodents [10]. In this study, we explored the intestinal microbes of five wild zokor species. It was indicated that although sharing similar compositions, the microbial proportions were distinct at both the phylum and genus levels. These zokor species exhibited distinct microbial diversity, structure, and predicted functions. In addition, we have discovered several biomarkers that can distinguish these zokor species.

Our results demonstrated that the intestinal microbes of zokors were in line with the gut microbial characteristics of herbivorous mammals. It was found that the intestinal microbial composition of five zokor species consisted of the same phyla as those of other rodents and mammals. Specifically, Firmicutes and Bacteroidota were the two dominant phyla in the gut microbes of mammals [30,50,51]. Furthermore, *E. cansus* and *E. smithi* have a higher ratio of Firmicutes/Bacteroidota compared to the other three species. Firmicutes were the most prevalent phylum within wild rodent microbiota when compared with laboratory-reared rodents [52,53]. Firmicutes and Bacteroidota were primarily accountable for food fermentation within the intestine [54], and a large proportion of Firmicutes/Bacteroidota was associated with a carbohydrate diet [55]. Therefore, the abundant Firmicutes and Bacteroidota in the zokor intestine enhance the decomposition of complex carbohydrates such as cellulose and hemicellulose, corresponding to the intestinal microbial characteristics of herbivores. This is similar to the findings in our previous studies [18]. At genus level, the intestinal microbes of five zokor species were predominantly dominated by *Muribaculaceae*, *Lachnospiraceae_NK4A136_group*, *unclassified_f_Lachnospiraceae*, and *uncultured_f_Lachnospiraceae*. This indicates the conservation of intestinal microbial composition throughout the evolution of rodents even mammals, featuring selective colonization of essential microbiota regardless of host identity [56]. In zokors, three dominant genera belong to the family Lachnospiraceae, which is associated with the breakdown of complex carbohydrates in plant materials [57,58]. *Muribaculaceae* might be related to the degradation of various complex carbohydrates [59]. The prevalence of these bacteria in the intestine of zokors is likely related to the plant-based diet of zokors. As a result, the intestinal microbial composition of zokors at genus level is also consistent with the gut microbial characteristics of herbivores.

Our results indicated that species’ identity, phylogeny, and geographical locations could impact the intestinal microbial composition of zokor. It is widely accepted that host genetics and ecology play an important role in shaping the intestinal microbes of wild mammals [30,31,60]. We have discovered that there exist differences in intestinal microbial diversity among five zokor species. *E. cansus* harbored more diverse microbes than that of other zokor species. Studies have shown that more diverse microbes were associated with the complex fibers present in plant roots [60], suggesting that the diversity of intestinal microbes in *E. cansus* may help them in digesting and obtaining energy from limited food resources. There was no significant difference in alpha diversity between *E. cansus* and *E. rothschildi*. It is likely due to the fact that the *E. rothschildi* samples were collected from different seasons (summer and autumn), as seasons have a certain impact on the diversity of intestinal microbes. In addition, we have found that zokor species harbored highly different microbes from one another (RERMANOVA test, F = 3.60, R^2^ = 0.1215, *p* = 0.001). Moreover, there was also a distinct structural separation of the intestinal microbes between zokors from *Eospalax* and *Myospalax* (RERMANOVA test, F = 6.74, R^2^ = 0.0593, *p* = 0.001). These results showed a close correlation between the microbial similarity and phylogenetic relationships of the hosts. Therefore, the hosts that were closely related had more similar intestinal microbes compared to divergent ones, which was similar to some findings of previous studies on other wild rodents [61,62,63,64,65]. Furthermore, we have found a significant geographical differentiation in the intestinal microbes of five zokor species (RERMANOVA test, F = 5.35, R^2^ = 0.0916, *p* = 0.001). Within the genus *Eospalax*, *E. cansus* was more closely related to *E. rothschildi* than to *E. smithi* [66,67]. However, the *E. cansus* and *E. smithi* collected in this study are sympatric distributed in the northwest of China, far from *E. rothschildi* which is distributed in the central part of China. In the PCoA plot (Figure 3), *E. cansus* and *E. smithi* had more similar microbes than those of *E. rothschildi*, thereby demonstrating the significance of the geographical location and ecological environment of the habitat for the formation of intestinal microbes in zokors. Moreover, *M. aspalax* and *M. psilurus*, who live in the Eurasian steppe zone of the northeast of China, are also nearly sympatric and closely related [19]. These differences related to host genetics and ecology may play an important role in shaping the intestinal microbes of zokors.

Our study also uncovered certain differences in microbial composition and function among five zokor species. LefSe analysis was used to screen biomarkers that significantly differ among five zokor species. The genera enriched in *E. cansus* and *E. smithi* may be linked to the degradation of complex carbohydrates, potentially serving as an adaptation to limited food resources underground [18]. Most of the species of *Lactobacillus* were characterized as probiotics and can utilize a wide variety of carbohydrates as substrates for fermentation [68]. *Ruminococcus* was a typical cellulose-decomposing bacterium [69]. *Unclassified_f_Lachnospiraceae* and *Lachnospiraceae_NK4A136_group* both belong to Lachnospiraceae, and they are capable of degrading the complex plant bran of recalcitrant substrate [57,58]. The genera enriched in *E. rothschildi* were associated with the catabolism of various ingredients, which might be an adaptation to the abundant food resources in its habitat [18]. *Muribaculaceae* may be connected with the degradation of a variety of carbohydrates [59]. *Uncultured_f_Christensenellaceae* belongs to the family Christensenellaceae, which has numerous functions such as the catabolism of protein and prebiotic fibers [70,71,72,73]. Moreover, the genera enriched in *M. aspalax* and *M. psilurus* were also related to the metabolism of various substances, which might be in line with their diverse food resources [19]. *Unclassified_f_Christensenellaceae* can degrade protein and fibers [70,71,72,73]. Species of *Desulfovibrio* were groups of sulfate-reducing bacteria that could decompose sulfate into hydrogen sulfide [74]. *Uncultured_f_Lachnospiraceae* had the capacity to decompose the complex plant bran of recalcitrant substrate [57,58]. *Unclassified_f_Oscillospiraceae* could effectively utilize the carbohydrates and monosaccharides of tender leaves and roots [75]. Furthermore, the functional prediction of intestinal microbes revealed distinct functional spectra among five zokor species. There are significantly differential functions at KEGG level 2 under the metabolism function. The differences in metabolic capacities of intestinal microbes might have an impact on their capacity to carry out specific metabolic functions, modulate host physiology, and react to environmental changes [76]. *E. cansus* and *E. smithi* showed gene functional enrichment associated with carbohydrate metabolism and amino acid metabolism. This serves as an indication of a focus on nutrient acquisition and processing, corresponding to obtaining maximum energy from the limited ingredients in plant roots [56]. Conversely, *E. rothschildi*, *M. aspalax*, and *M. psilurus* showed gene functional enrichment related to cofactors and vitamins metabolism, and glycan biosynthesis and metabolism, indicating their metabolic capacity for diverse substances.

In this study, the intestinal microbes of five zokor species were investigated. It is shown that the intestinal microbes of five zokor species are similar in composition but distinct in proportions. In addition, these zokor species exhibited distinct microbial diversity and structure. This study is subject to certain limitations. Due to the differences in zokor population and sampling challenges, sample size imbalance occurred across different species. In particular, some of the zokor samples were collected in different seasons (*E. rothschildi*), and additionally, some samples (*E. smithi*) were not sourced from their typical distribution regions. Through data analysis, we deduced that these limitations did not significantly affect the primary conclusion of the study. To ensure greater reliability of the research outcomes, in subsequent research, we planned to enlarge the sample size and gather more samples from the typical distribution regions of each zokor species within the same season. Furthermore, we intended to isolate cellulose-degrading bacteria like *Ruminococcus* from the zokors. These bacteria can improve soil quality and remediate polluted soil, as well as enhance the intestinal microenvironment of animals, increase feed utilization, and contribute to better animal production performance. Additionally, we will also isolate probiotic bacteria like *Lactobacillus*. These probiotics have the capacity to balance the intestinal microbes, improve the intestinal microenvironment and enhance the host’s immune function.

## Figures and Tables

**Figure 1 microorganisms-13-00027-f001:**
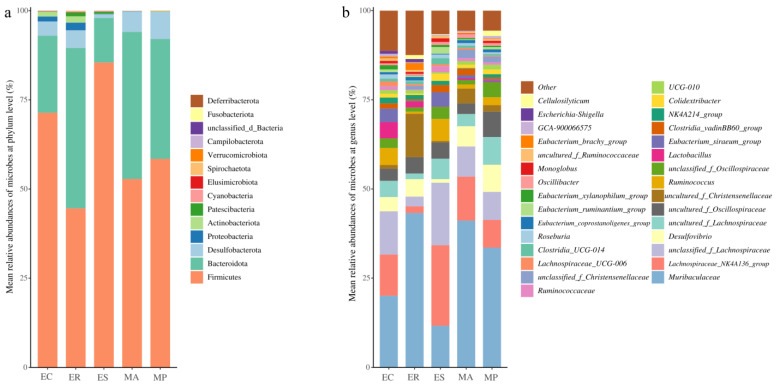
Mean relative abundances of microbes (**a**) at phylum level and (**b**) at genus level across all zokor samples.

**Figure 2 microorganisms-13-00027-f002:**
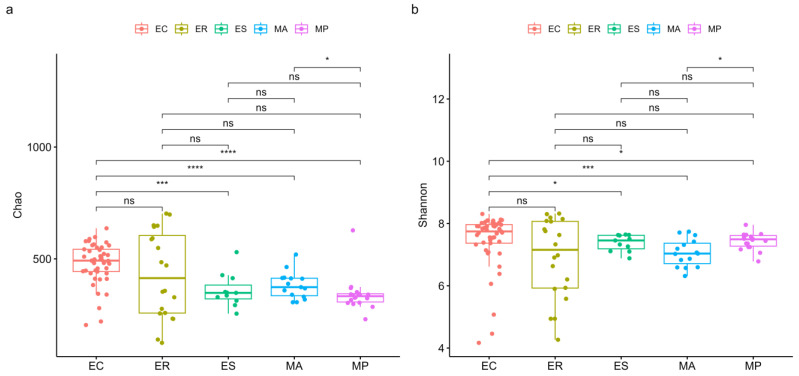
The comparisons of alpha diversity of intestinal microbes among five zokor species. (**a**) Chao index. (**b**) Shannon index. Asterisks indicate significant differences between two species (* *p* < 0.05, *** *p* < 0.001, **** *p* < 0.0001); ns indicates no significant differences between two species (ns: *p* ≥ 0.05).

**Figure 3 microorganisms-13-00027-f003:**
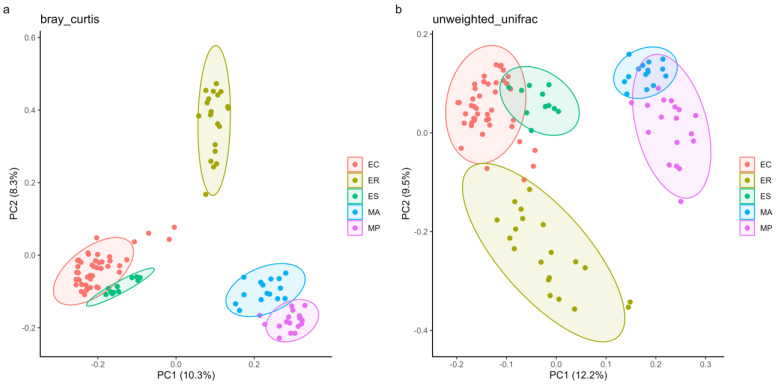
Principal coordinates analysis (PCoA) of bacterial communities of five zokor species across all samples based on (**a**) Bray–Curtis and (**b**) Unweighted–Unifrac distance.

**Figure 4 microorganisms-13-00027-f004:**
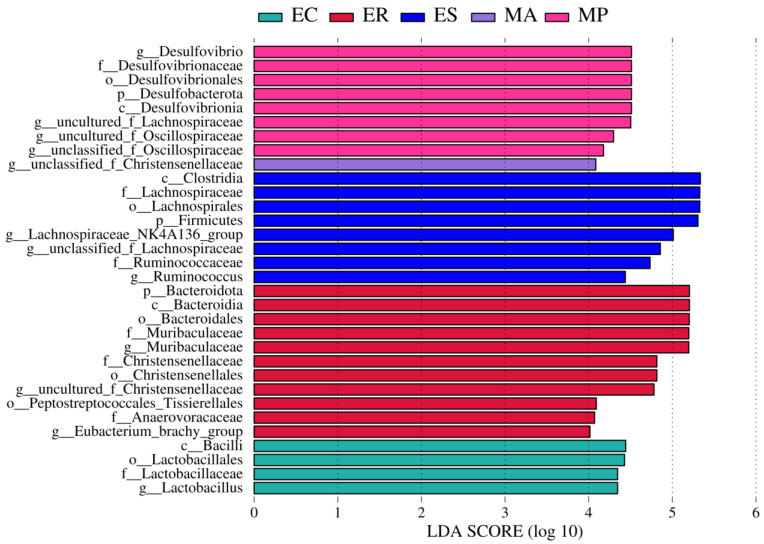
Histograms showing enriched taxa for each zokor species based on linear discriminant analysis effect size (LefSe analysis) (LDA score > 4.0).

**Figure 5 microorganisms-13-00027-f005:**
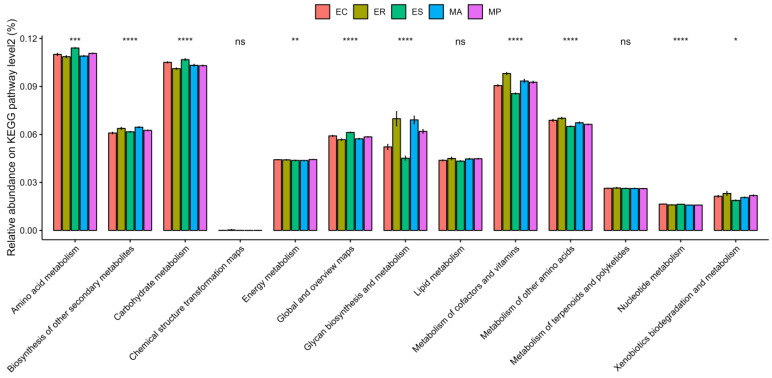
The comparison of predicted functional differences in intestinal microbes among five zokor species under metabolism function of KEGG pathway level 2. Asterisks indicate significant differences among species (* *p* < 0.05, ** *p* < 0.01, *** *p* < 0.001, **** *p* < 0.0001); ns indicates no significant differences among species (ns: *p* ≥ 0.05).

## Data Availability

Raw sequencing data were deposited in the NCBI Sequence Read Archive (SRA) database under the BioProject PRJNA664217, PRJNA833598 and PRJNA1130553.

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
