# Peer review of "Investigation of Intestinal Microbes of Five Zokor Species Based on 16S rRNA Sequences"

_microorganisms, 2024, doi:10.3390/microorganisms13010027_

Round 1
Reviewer 1 Report
Comments and Suggestions for Authors
The article titled "Investigation of intestinal microbes of five zokor species based on 16S rRNA sequences" which used 16S rRNA sequencing for characterizing microbial diversity in five different species of Zokor rodent. The manuscript is written well, however, authors are requested to revise the manuscript based on the following comments.
1- In the abstract, it is overly result-focused and lacks methodology details, so a brief mention of the methodology framework in the right place of the abstract can improve the readability of the study, so it recommended to revise the abstract and include the sentences summarizing the methodology before mentioning the results. In addition to that the significance of the results should be clearly mentioned.
2- In introduction, the authors provide a general overview of the intestinal microbiota as zokors rodent and the ecological and physical adaptation in their environment, but this part contains a redundancies and lack of depth into a specific gap the study aims to manage. So, it would be better to include what is known currently about microbiota in Zokor rodent.
3- State the limitations of the study especially the huge differences in the sample size between the groups, and the sampling location, is that influence the microbial diversity and accordingly the results?
4- In discussion section the authors discussed the metabolic functions and ecological behaviors, it would be beneficial to discuss how these metabolic functions can be translated to ecological benefits and advantages.
5- As stated in the manuscript that the sampling time was between 2019-2021, what about the seasonal variations is that affect the microbial diversity and as a result influence the results?
Comments on the Quality of English Language
Minor editing is required
Reviewer 2 Report
Comments and Suggestions for Authors
Line 69: What is the purpose of studying the microbiomes of Zokors? Is the Zokor considered damaging to the ecosystem or a potential threat to human health?
Line 74: Were three species or five species included in the study?
Confounding Factor: Since two species were sampled from one region and one from a different region, the effect of species appears to be confounded with the biogeographic factor in this design. Could this be clarified?
Age and Gender Distribution: How were the age and gender distributions between the species balanced?
Line 103: For the two additional species, what were their biogeographic details, and how were age and gender distributed?
Sequencing Variation: What steps were taken to minimize variation between sequencing runs?
Line 141: Was the IACUC protocol number provided for this study?
Line 198: Could you clarify the comparison between EC versus ES and EC versus ER?
Line 225: Was the observed difference due to the genetic background of each species, or could it be attributed to biogeographic location?
Line 254: Perhaps it would be more accurate to refer to this as "genetic background"?
Line 301: The relatively high abundance of certain metabolic pathways makes it difficult to differentiate low-abundance pathways. The authors may consider moving those not described in the text to the supplementary materials.
